# Oxidative Stress and Mitochondrial Dysfunction in Myelodysplastic Syndrome: Roles in Development, Diagnosis, Prognosis, and Treatment

**DOI:** 10.3390/ijms26136415

**Published:** 2025-07-03

**Authors:** Federico Pierro, Manlio Fazio, Giuseppe Murdaca, Fabio Stagno, Sebastiano Gangemi, Alessandro Allegra

**Affiliations:** 1Division of Hematology, Department of Human Pathology in Adulthood and Childhood “Gaetano Barresi”, University of Messina, Via Consolare Valeria, 98125 Messina, Italy; federico.pierro@studenti.unime.it (F.P.); manliofazio@hotmail.it (M.F.); fabio.stagno@unime.it (F.S.); aallegra@unime.it (A.A.); 2Department of Internal Medicine, University of Genova, 16126 Genova, Italy; 3Allergology and Clinical Immunology, San Bartolomeo Hospital, 19038 Sarzana, Italy; 4Allergy and Clinical Immunology Unit, Department of Clinical and Experimental Medicine, University of Messina, Via Consolare Valeria, 98125 Messina, Italy; gangemis@unime.it

**Keywords:** myelodysplastic syndromes, oxidative stress, reactive oxygen species, iron overload, hematopoietic stem cells, DNA damage, epigenetic regulation, mitochondrial dysfunction, redox-targeted therapy

## Abstract

Myelodysplastic syndromes are a group of clonal hematopoietic stem cell disorders characterized by ineffective hematopoiesis, peripheral cytopenia, and dysplasia in one or more myeloid lineages, with a variable risk of progression to acute myeloid leukemia. In addition to well-characterized genetic and epigenetic abnormalities, oxidative stress has emerged as a critical contributor to the pathophysiology of myelodysplastic syndrome. Reactive oxygen species and reactive nitrogen species can induce cumulative DNA damage, mitochondrial dysfunction, and altered redox homeostasis, promoting genomic instability and clonal evolution. Elevated oxidative stress in patients with myelodysplastic syndromes has been linked to increased apoptosis of hematopoietic stem and progenitor cells, disruption of the bone marrow microenvironment, and progression toward leukemic transformation. Moreover, ROS-related pathways, such as TP53 mutations and epigenetic dysregulation, interact with the key molecular drivers of myelodysplastic syndrome. Given these findings, oxidative stress is now recognized not only as a hallmark of disease biology but also as a potential therapeutic target. Antioxidant-based strategies and agents that modulate redox signaling are being investigated for their ability to restore hematopoietic function and enhance treatment efficacy. This review provides an overview of the current biology of myelodysplastic syndrome, highlights the connections between oxidative stress and disease mechanisms, and explores emerging redox-targeted therapeutic approaches.

## 1. Introduction

### General Considerations for Myelodysplastic Syndromes

Myelodysplastic syndromes (MDS) are hematological disorders characterized by peripheral cytopenias and hypercellular marrow, mainly affecting the elderly, and possibly progressing to acute myeloid leukemia (AML) through clonal evolution [1,2].

Over time, the classification of MDS has evolved. To enhance diagnostic accuracy, the 2016 WHO Classification incorporated molecular data into MDS diagnosis and management [3], notably identifying MDS with ring sideroblasts (MDS-RS) by SF3B1 mutations and ≥5% ring sideroblasts. Isolated del(5q) is a distinct subtype with good prognosis and response to lenalidomide, whereas TP53 mutations have poor outcomes [3]. Building on this, the 2022 WHO Classification redefined MDS into two main categories: genetically defined (MDS-5q, MDS-SF3B1, and MDS-biTP53) and morphologically defined (MDS with Low Blasts, MDS with Hypocellularity, and MDS with Increased Blasts) [4]. Key updates include reclassifying NPM1-mutated cases as AML and former MDS-U cases as Clonal Cytopenia of Undetermined Significance (CCUS) [5], underscoring the growing importance of genetic profiling in disease categorization.

However, differentiating MDS from related conditions remains challenging owing to overlapping features, making genetic analysis essential for prognosis and treatment decisions [6]. A variety of genetic alterations, including cytogenetic abnormalities, somatic mutations, and gene expression changes, drive MDS pathogenesis [7]. Frequent abnormalities have included del(5q), del(7q), trisomy 8, and complex karyotypes [7]. In addition, promoter hypermethylation and mutations in genes that regulate DNA methylation and histone modifications are commonly observed [7]. Particularly relevant are mutations in RNA splicing genes (*SF3B1*, *SRSF2*, *U2AF1*, and *ZRSR2*) and epigenetic regulators (TET2, ASXL1, and DNMT3A); spliceosome mutations are present in approximately 50% of cases [8,9]. Mutations affecting signal transduction pathways (e.g., NRAS and JAK2) and transcriptional regulators (e.g., RUNX1 and TP53) are also frequently encountered [8,9].

Gene expression profiling has contributed to the identification of dysregulated genes, and altered signaling pathways and prognostic molecular signatures [8]. Moreover, mutations in genes such as *DNMT3A*, *ASXL1*, and *TET2* may predict better responses to hypomethylating agents, supporting the integration of genomic, transcriptional, and clinical data to guide patient management [8].

Furthermore, in a therapeutical context, mutations in SF3B1 and IDH1/2 provide opportunities for targeted interventions. Consequently, molecular profiling using NGS has become one of the cornerstones of diagnosis and therapeutic planning for MDS [10].

In the context of pathogenesis, beyond genetic alterations, immune dysregulation plays a key role in MDS pathogenesis; it is often described as the “Yin–Yang theory” of immune activation and suppression, and is characterized by immune hyperactivity and increased apoptosis in low-risk MDS (LR-MDS), while high-risk MDS (HR-MDS) exhibits immune suppression [11].

Both innate and adaptive immune responses are hyperactive in the bone marrow of MDS patients, and autoimmune diseases are frequently observed, indicating a strong immunological component [12].

Supporting this, a 2012 study described autoimmune manifestations (AIMs) in some MDS patients that contributed to ineffective hematopoiesis via immune-mediated attacks on hematopoietic clones [13].

Therapeutic approaches often involve the use of steroids or immunosuppressants. Notably, azacytidine has shown potential in reducing steroid dependence in select cases [14].

## 2. Risk Stratification and Treatment Options

### Prognostic Tools and Standard Treatment Approaches

Building on these diagnostic and pathogenic considerations, treatment decisions for MDS are strongly influenced by risk stratification tools.

The Revised International Prognostic Scoring System (IPSS-R), introduced in 2012, improves prognostic accuracy by adding new risk groups, refining blast and cytopenia evaluation, and classifying patients into five risk levels; it has improved prognostic precision by refining blast and cytopenia evaluation and dividing patients into five risk categories [15]. Additional prognostic indicators such as age, performance status, and ferritin and LDH levels have been associated with survival; although correlated with survival, they are not necessarily correlated with AML transformation [15]. More recently, Molecular IPSS (IPSS-M) has incorporated mutational data from mutations in over 30 genes (e.g., *TP53*, *ASXL1*, *RUNX1*, and *TET2*) to better predict survival, AML progression and disease evolution [16].

These tools reflect the growing role of next-generation sequencing (NGS) in MDS diagnosis, classification, prognosis, and treatment selection [17]. Treatment strategies are risk-adapted, and guided by IPSS-R or IPSS-M scores [18].

In low-risk MDS (LR-MDS), marked by dysplasia, low blast counts, and favorable genetics, therapy is largely supportive [19]. This includes transfusions, growth factors, and approved agents such as lenalidomide, HMAs (azacitidine, decitabine), and luspatercept. However, HSCT (hematopoietic stem cell transplantation) remains underutilized, despite curative potential [19,20].

Erythropoiesis-stimulating agents (ESAs), including EPO and darbepoetin alfa, are first-line treatments for anemia in patients with EPO < 500 U/L and low transfusion needs, improving hemoglobin and reducing transfusion burden [21].

Luspatercept increases transfusion independence in MDS-RS and is effective in ESAs-refractory cases [22].

Lenalidomide is particularly effective in del(5q) MDS, with 67% achieving transfusion independence and ~50% cytogenetic response [23]. Epoetin-α p shows erythroid responses in 32–46% of patients, far exceeding the percentages for placebo [24].

In contrast, in high-risk MDS (HR-MDS), the goal is to delay AML progression and improve survival. Standard care includes azacitidine and decitabine [25]. Notably, azacitidine improves survival in patients with chromosome 7 abnormalities [25].

The oral formulation of CC-486 (oral azacitidine) has also attracted attention. While it enhances transfusion independence and bilineage response in LR-MDS, it has also been associated with reports of increased early infection-related mortality [26]. In elderly AML patients, CC-486 extends survival by 10 months and reduces relapse [27].

Given the limitations of parenteral HMAs, oral formulations improve convenience and patient adherence [28,29].

Meanwhile, the bone marrow microenvironment (BME) in MDS supports malignant clones over healthy hematopoiesis. Azanucleotides may affect both malignant and stromal components; lenalidomide benefits patients with specific deletions, while luspatercept enhances erythropoiesis without clonal expansion [30].

Finally, the immunosuppressive BME also facilitates malignant persistence. Myeloid-derived suppressor cells (MDSCs), often expanded in MDS, correlate with progression and induce dysplasia in models, making them promising therapeutic targets [31,32].

## 3. New Treatment Approaches

### Overview of Innovative Therapies Under Clinical Development

New agents such as roxadustat, imetelstat, oral HMAs, TP53 modulators, and magrolimab are being developed to overcome current treatment limitations and address molecular targets in MDS [21].

For example, splicing modulators aim to correct the common splicing defects in MDS. Inhibition of CHRM4 in early erythroid progenitors has emerged as a novel strategy with encouraging preclinical results. Splicing modulators and CHRM4 inhibitors in early erythroid progenitors also show promise [33].

Roxadustat, an oral Hypoxia-Inducible Factor–Prolyl Hydroxylase (HIF-PH) inhibitor, represents a mechanistically distinct approach to treating anemia, compared with ESAs. Clinical trials have demonstrated that it achieved 37.5% transfusion independence at both 28 and 52 weeks, with over 50% of patients experiencing a ≥50% reduction in transfusion requirements, particularly at a dose of 2.5 mg/kg [34].

Additionally, roxadustat has shown efficacy in improving anemia and lowering transfusion dependency in MDS patients with concurrent chronic kidney disease [35].

Roxadustat, an oral HIF-PH inhibitor, differs mechanistically from ESAs. It achieved 37.5% transfusion independence at both 28 and 52 weeks, with over half of patients reducing transfusion needs ≥50%, especially at 2.5 mg/kg [34]. It has also shown benefits in MDS patients with chronic kidney disease [35].

Imetelstat, a telomerase inhibitor, interferes with telomere maintenance by targeting the RNA component hTR, effectively impairing malignant hematopoietic stem cells (HSCs), hematopoietic progenitor cells (HPCs), and leukemic stem cells (LSCs), while sparing normal cells [36]. Currently approved for LR-MDS patients with transfusion-dependent anemia, imetelstat (7.1 mg/kg every 4 weeks) demonstrated improved red blood cell transfusion independence at 8 and 24 weeks, compared to placebo [36].

Targeting immune checkpoints is an emerging strategy. CD47, a key innate immune checkpoint, has become a compelling therapeutic target for MDS and AML, while magrolimab, an anti-CD47 monoclonal antibody, has shown synergistic effects when combined with azacitidine, particularly in high-risk patients with TP53 mutations [37].

Finally, sabatolimab, an anti-TIM-3 antibody, has also demonstrated durable responses when administered alongside HMAs in HR-MDS and AML patients. One study reported a complete remission (CR) rate of 33% and an overall response rate (ORR) of 75%, indicating robust clinical efficacy; additionally, it shows efficacy in ESA-refractory cases [36,38,39].

This review presents an overview of the contemporary biology of myelodysplastic syndrome, emphasizes the relationships between oxidative stress and disease processes, and examines novel redox-targeted therapy strategies.

## 4. Oxidative Stress: Mechanisms and Disease Connections

### 4.1. Oxidative Stress Plays a Role in Both Physiological and Pathological Contexts

Reactive oxygen species (ROS) and reactive nitrogen species (RNS) are byproducts of normal cellular metabolism and serve dual roles as damaging agents and essential signaling molecules. While excess ROS/RNS levels induce oxidative and nitrosative stress, damaging DNA, proteins, and lipids, physiological levels are crucial for redox homeostasis, immune responses, and intracellular signaling, as discussed by Valko et al. [40].

Oxidative stress has been implicated in a wide range of diseases, including cancer, cardiovascular and neurodegenerative disorders, diabetes, and aging [40]. ROS-induced damage contributes to chronic disease progression via lipid peroxidation, protein oxidation, and DNA strand breaks, as shown in studies by Cabello-Verrugio et al. and Kudryavtseva et al. [41,42]. Moreover ROS-induced cellular senescence may act as a tumor-suppressive barrier. However, the senescence-associated secretory phenotype (SASP) can paradoxically promote tumorigenesis. Targeting the key pathways involved in this process may suppress malignant progression, as suggested by Li et al. [43].

ROS and RNS regulate autophagy, particularly during nutrient deprivation. The p62/Keap1/Nrf2 axis plays a leading role in the relationship between oxidative stress and antioxidant response. Autophagy influences DNA repair mechanisms, with ROS and RNS functioning as upstream modulators that determine cellular fate under stressful conditions, according to Filomeni et al. [44].

Ferroptosis, a regulated, iron-dependent form of cell death marked by lipid ROS accumulation, plays a role in neurodegenerative diseases, such as Alzheimer’s and Parkinson’s [45]. Recent studies have explored redox regulation and ferroptosis in cancer, suggesting that targeting these pathways may provide therapeutic benefits, especially in resistant or metastatic tumors, as highlighted by Zeng et al. [46].

Altered energy metabolism has also been associated with oxidative stress. Dysregulated glycolysis and oxidative phosphorylation (OXPHOS) have been observed in several diseases. For instance, Jourdain et al. [47] showed that loss of LUC7L2, a component of the pre-mRNA splicing machinery, leads to enhanced OXPHOS and reduced glycolysis; this finding links aberrant splicing to oxidative stress and disease.

Mitochondrial dysfunction and ROS are key drivers of inflammatory myopathies such as inclusion body myositis (IBM), as reported by Danieli et al. [48]. Similarly, in muscular dystrophies (MDs), intracellular ROS accumulation causes contractile dysfunction. Monoamine oxidase (MAO), a major ROS source, is upregulated in dystrophic muscles, and its inhibition with pargyline shows therapeutic promise, as described by Canton et al. [49].

In the liver, oxidative stress plays a key role in hepatocellular injury, particularly during hypercholesterolemia, which increases ROS levels and apoptosis, according to Jadeja et al. [50].

Oxidative stress is a key factor in osteoporosis, especially in high-risk groups (e.g., postmenopausal women and patients with metabolic disorders). Oxidative stress disrupts bone remodeling, and mitochondria-targeted antioxidants are currently being investigated. Vitamin D deficiency worsens oxidative damage and is linked to osteoporosis, autoimmune diseases, and hematologic malignancies [51,52].

In cardiovascular diseases, oxidative stress impairs endothelial function, contributing to atherosclerosis, arrhythmias, and hypertension. As noted by Senoner and Dichtl [53], a Mediterranean diet rich in natural antioxidants reduces oxidative damage and cardiovascular risk.

### 4.2. Oxidative Stress and Mitochondrial Dysfunction in Cancer Biology and Therapy Response

In oncology, oxidative stress is an established mediator of both tumorigenesis and the treatment response. As described by Yamada et al. [54], various therapeutic strategies for regulating mitochondrial oxidative stress have been developed, including photodynamic therapy (PDT), photothermal therapy (PTT), and antioxidant therapy using a mitochondrial drug delivery system (DDS). The PDT effect arises from the dynamic interaction between a photosensitizer (PS), light, and molecular oxygen, resulting in the generation of lethal levels of ROS, primarily highly reactive singlet oxygen [54].

For example, PDT uses controlled oxidative damage to target skin cancer cells and is more effective than chemotherapy or surgery, as reported by Allegra et al. [55]. Cancer cells display the Warburg effect, relying on glycolysis despite oxygen availability, and they typically have high ROS levels. Their survival depends on enhanced antioxidant defenses, which can be selectively targeted for therapeutic purposes, as highlighted by Gorrini et al. [56].

A recent study by Allegra et al. [57] highlighted significant sex differences in cancer susceptibility, with males generally experiencing higher oxidative stress and cancer incidence than females. Young women are more protected from cancer due to hormonal regulation of the redox state and mitochondrial function. Gender differences in immunological responses and antioxidant enzyme activity influence cancer susceptibility and treatment outcomes.

ROS signaling also plays a key role in the regulation of mesenchymal stem cells (MSCs) by controlling their proliferation and differentiation through pathways such as p38-MAPK and glycolysis. SIRT1, a part of the NAD/SIRT axis, helps maintain stem cell function, and its loss accelerates aging phenotypes [58].

Cancer stem cells (CSCs) are particularly resilient, owing to their efficient ROS scavenging and DNA repair mechanisms. These features make them highly resistant to chemotherapy and radiotherapy [59].

In neoplastic myeloid diseases, including MDS, elevated ROS levels contribute to clonal evolution and progression. As shown by Hole et al. [60], the targeting of ROS pathways has dual potential: antioxidant approaches may reduce leukemic proliferation and protect normal cells from chemotherapy-induced damage, whereas pro-oxidant therapies can exploit ROS overload to selectively induce leukemic cell death.

Oxidative stress markers are also elevated in patients with multiple myeloma (MM). Gangemi et al. [61] reported that serum levels of advanced oxidation protein products (AOPPs) and S-nitrosylated proteins are significantly higher in MM patients than in MGUS patients and healthy controls, especially those MM patients with bone lesions.

Similarly, Hodgkin’s lymphoma (HL) patients exhibit elevated levels of malondialdehyde (MDA), 4-HNE, and protein carbonyl groups, indicating increased lipid peroxidation and protein oxidation, compared to healthy individuals, as demonstrated by Morabito et al. [62]. These findings confirmed the role of oxidative stress in pathogenesis.

Further supporting the role of oxidative stress in cancer prognosis, Xie et al. [63] conducted a machine learning (ML) study on esophageal squamous cell carcinoma (ESCC) patients and found that oxidative stress markers, along with tumor and staging features, were among the strongest predictors of 3-year overall survival. The Random Forest (RF) model exhibited superior performance in both the training and validation sets.

On the other side, mitochondrial dysfunction is a common characteristic in cancer cells and contributes to cancer progression. Mitochondrial stress responses, including Mitochondrial Unfolded Protein Response (UPRmt) and Integrated Stress Response (ISR), play a role in cancer cell adaptation and growth, as discussed by Wang et al. [64].

A recent study by Winter et al. [65] identified the essential role of the electron transport chain (ETC) in generating aspartate in proliferating cells, which is a key finding in cancer cell metabolism. This finding describes how ETC dysfunction activates OMA1, a Zn^2^⁺-dependent metalloendopeptidase, in the inner membrane of mitochondria, leading to the cleavage of DELE1 and activation of ISR, which is crucial for understanding how mitochondrial stress signals apoptosis. Prolonged stress activates a pro-apoptotic response through ISR, which converges on BCL2 family proteins to trigger mitochondrial outer membrane permeabilization (MOMP).

Finally, mitochondrial dysfunctions due to mtDNA mutations, Tricarboxylic Acid (TCA) cycle enzyme defects, and oxidative stress contribute to cancer development by disrupting metabolic pathways and redox balance. As noted by Luo et al. [66], this can lead to the accumulation of oncometabolites (D-2-hydroxyglutarate and succinate), inhibition of prolyl hydroxylases, and HIF1α stabilization. Additionally, enhanced NOX activity, increased ROS generation, and altered redox status play central roles.

## 5. Oxidative Stress in Myelodysplastic Syndromes

### 5.1. Oxidative Mechanisms Specific to MDS Pathogenesis

In the context of MDS, the association between oxidative stress and disease pathogenesis has become increasingly evident over the past few decades. HSC function is influenced by both intrinsic and extrinsic elements, with hypoxia playing a vital role in maintaining quiescence and metabolic dormancy. Leukemic stem cells exhibit distinct metabolic traits, relying more heavily on oxidative respiration and being more susceptible to oxidative stress than are normal HSCs. In AML cells, mitochondrial abnormalities, such as a shift to aerobic glycolysis, offer potential for therapeutic strategies, particularly through the use of IDH2 inhibitors, as discussed by Testa et al. [67].

Human hematopoietic progenitors and stem cells (HPSCs) exhibit a metabolic phenotype characterized by high glycolysis rates and low oxygen consumption. Cells with a low mitochondrial potential are enriched in HPSCs and have better repopulation abilities. The metabolic phenotype of HPSCs is regulated by Meis1 and its cofactors Pbx1 and HoxA9 through the transcriptional activation of Hif-1α, as shown by Kocabas et al. [68]. Moreover, a recent study revealed that in the initial phases of human red blood cell development, there is an increase in glutaminolysis and the production of α-ketoglutarate (αKG), which propels mitochondrial OXPHOS [68].

Furthermore, Gonzalez-Menendez et al. [69] highlighted that isocitrate dehydrogenase 1 (IDH1) is pivotal in managing the conversion between αKG and isocitrate, a process essential for the later stages of erythropoiesis. IDH mutations and their implications in cancer development have been reported previously. Specific IDH inhibitors, such as enasidenib and ivosidenib, have been approved and used in therapy [70].

Hypoxia-inducible factor 1-alpha (HIF-1α) is a central transcription factor that orchestrates the cellular adaptation to low oxygen availability [71]. Under normal oxygen conditions, HIF-1α is hydroxylated on conserved proline residues by prolyl hydroxylase domain (PHD) enzymes, which facilitates its recognition by the von Hippel–Lindau (VHL) E3 ubiquitin ligase complex, leading to proteasomal degradation, as first described by Jaakkola et al. [72]. In hypoxia, oxygen scarcity inhibits PHD activity, allowing HIF-1α to escape degradation, accumulate in the cytoplasm, translocate into the nucleus, and dimerize with HIF-1β. This heterodimer binds hypoxia response elements (HREs) in the promoter regions of target genes, activating transcriptional programs involved in glycolysis, angiogenesis (e.g., VEGF), and cell survival [73].

Importantly, these pathways have been implicated in the maintenance of leukemic stem cells and in the progression of hematologic malignancies such as MDS and AML, as reviewed by Deynoux et al. [74].

Beyond its role in oxygen sensing, HIF signaling exerts multifaceted effects on leukemogenesis by regulating energy metabolism, cell-cycle control, and immune modulation. The role of HIFs in leukemia is highly context-dependent, with evidence indicating that both oncogenic and tumor-suppressive activities are influenced by disease stage, microenvironmental factors, and genetic background [74].

Functional studies across various leukemia subtypes have shown that HIF overexpression promotes bone marrow neoangiogenesis, sustains LSC self-renewal, and contributes to chemotherapy resistance. Magliulo and Bernardi [75] further emphasized that HIF-1α has emerged as a critical regulator of LSC maintenance and dissemination, underscoring its role in leukemic propagation. In the context of MDS, aberrant activation of HIF-1 and pseudohypoxia signaling constitutes a hallmark of disease pathophysiology. These alterations are not merely bystanders, but actively contribute to ineffective hematopoiesis and disease progression. Notably, HIF-1α activity appears to correlate with disease severity, with therapeutic efficacy primarily in patients with lower blast counts, as noted by Stergiou et al. [76].

Disruption of the bone marrow microenvironment further exacerbates oxidative stress, leading to premature hematopoietic stem cell exhaustion or malignant transformation. The bone marrow microenvironment, particularly the endosteal niche, is crucial for maintaining quiescent hematopoietic stem cells through hypoxia. In hematologic malignancies, the bone marrow is highly hypoxic, and this hypoxia is associated with disease progression and stabilization of oncogenic factors like HIF-1α [77].

Moreover, constitutive HIF-1α activation has been linked to multiple genetic lesions in MDS and has been shown to be sufficient to initiate MDS-like phenotypes in murine models. According to Chen and Steidl [78], both the genetic and pharmacologic inhibition of HIF-1α suppress disease development while sparing normal hematopoietic function.

These data suggest that dysregulated HIF-1α signaling may actively drive clonal selection and leukemic progression by altering the metabolic and redox state of hematopoietic progenitors. In particular, HIF-1α promotes a shift toward glycolytic metabolism and suppresses mitochondrial respiration, thereby reducing the efficiency of DNA repair mechanisms and increasing the mutational burden under oxidative stress. This pseudohypoxic environment facilitates the expansion of preleukemic clones over healthy hematopoiesis and may synergize with additional oncogenic mutations to accelerate disease evolution from MDS to AML. Consequently, targeting HIF-1α or its downstream metabolic programs may represent a promising strategy for preventing or delaying malignant transformation in HR-MDS.

The evaluation of oxidative stress levels in MDS patients can be a useful tool. In fact, in a 2011 study, Saigo et al. [79] reported that a significant majority (85%) of MDS patients had higher oxidative stress levels than the reference range, with a positive correlation between ROS and serum ferritin levels and a negative correlation between ROS and hemoglobin levels. In addition, both iron accumulation and severe anemia contribute to oxidative stress in MDS patients.

Serological and molecular markers of oxidative stress are frequently observed in MDS patients. These include elevated levels of malondialdehyde, a lipid peroxidation byproduct, as well as increased oxidative DNA damage, such as the presence of oxidized bases in CD34+ hematopoietic progenitor cells [80].

The underlying mechanisms contributing to oxidative stress in MDS are multifactorial and include mitochondrial dysfunction, which is often driven by iron overload, mitochondrial DNA mutations, systemic inflammation, and abnormalities in the bone marrow stromal microenvironment [80].

In a 2014 study, Chung et al. [81] detected increased ROS levels in bone marrow nucleated cells from NHD13 mice, a transgenic mouse model that expresses a fusion gene consisting of NUP98 and HOXD13, which recapitulates key features of human MDS, including ineffective hematopoiesis, multilineage dysplasia, and progression to AML. These alterations were associated with increased DNA double-strand breaks and G2/M phase cell-cycle checkpoint activation. In addition, the study found an increased mutation frequency in NHD13 BMNC, suggesting that oxidative stress contributes to MDS progression to AML through ineffective DNA repair and oncogenic mutations. Mechanistically, the NUP98-HOXD13 fusion impairs genomic stability by altering the expression of genes involved in DNA repair and oxidative homeostasis, thereby promoting the accumulation of double-strand breaks. This is further exacerbated by mitochondrial dysfunction, as NHD13 cells exhibit elevated mitochondrial ROS and reduced antioxidant buffering capacity, linking fusion-induced transcriptional deregulation to redox imbalance and clonal evolution.

Furthermore, Gonçalves et al. [82] reported that MDS patients have lower glutathione (GSH) content and higher levels of peroxide and superoxide than controls, indicating increased oxidative stress. The correlation between GSH levels and transfusion dependency, as well as between peroxide levels and serum ferritin levels, indicates the potential clinical implications of oxidative stress.

Oxidative stress levels vary by MDS subtype and risk group, suggesting a complex relationship between oxidative stress and disease progression [82].

Novotna et al. [83] observed that patients with low-risk MDS exhibited a significant increase in oxidative DNA damage, compared to control subjects. This increase was observed in both erythroid and non-erythroid bone-marrow-cell populations, suggesting that oxidative DNA damage contributes to genomic instability and disease progression in low-risk MDS.

Additionally, Ghoti et al. [84] found that low-risk MDS patients have higher ROS and lower GSH levels in RBC and platelets, compared to standard controls, and there is a correlation between serum ferritin levels and ROS in RBC and platelets, suggesting that iron overload generates oxidative stress.

Finally, Jing et al. [85] highlighted that the NADPH oxidase (NOX) family is also involved in the pathogenesis of MDS, suggesting potential therapeutic targets. In a 2009 study, McCrann et al. [86] found that high expressions of Nox1 and weak expression of Nox4 were present in megakaryocytes, in addition to no significant expression of Nox2. The inhibition of Nox enzymes results in the accumulation of low-ploidy megakaryocytes and the reduction of high-ploidy cells. In addition, downregulation of cyclin E and D3 was observed with NOX inhibition, and upregulation of cyclin E partially restored ploidy levels. (Table 1).

### 5.2. The Dual Role of Reactive Oxygen Species in the Pathogenesis of Myelodysplastic Syndromes

ROS exhibit a concentration- and context-dependent dual role in hematologic malignancies, including in MDS and acute AML. At low to moderate levels, ROS, particularly hydrogen peroxide (H_2_O_2_), function as essential second messengers that modulate signaling pathways involved in stem cell maintenance, proliferation, and survival. In both normal hematopoietic stem cells and leukemic stem cells, basal ROS levels regulate the activity of redox-sensitive transcription factors such as FOXO, HIF-1α, and NF-κB, sustaining quiescence and self-renewal capacity [58,87]. These mechanisms are hijacked by LSCs to promote resistance to apoptosis and chemotherapeutic agents, often through upregulation of antioxidant systems including NRF2, glutathione, and thioredoxin [88,89].

In contrast, elevated ROS levels, resulting from mitochondrial dysfunction, iron overload, inflammation, or oncogenic signaling, induce macromolecular damage that contributes to genomic instability and clonal evolution. DNA strand breaks, base modifications, lipid peroxidation, and protein oxidation have all been linked to disease progression in MDS and the transformation to AML [90,91]. Importantly, while LSCs maintain redox homeostasis under basal ROS levels, they become vulnerable when oxidative stress exceeds their buffering capacity. Emerging therapeutic approaches exploit this vulnerability by inducing oxidative overload or disrupting antioxidant defenses. For instance, ferroptosis-inducing compounds such as erastin and RSL3, as well as NRF2 pathway inhibitors, have shown selective cytotoxicity against leukemic cells in preclinical models [92,93].

These observations help explain the limited clinical success of broad-spectrum antioxidant therapies, which may indiscriminately suppress both damaging and physiologically beneficial ROS. Such non-specific interventions can impair normal hematopoietic function and blunt redox-dependent tumor-suppressive responses. A more effective strategy involves stratifying patients according to redox status and tailoring interventions accordingly, supporting antioxidant capacity in LR-MDS and addressing iron overload in MDS, while applying targeted pro-oxidant or redox-disruptive agents in HR-MDS or AML to selectively eliminate LSCs [94,95].

### 5.3. Impacts of Iron Dysregulation and Chelation Therapy on Oxidative Stress in Myelodysplastic Syndrome

Dysregulation of iron homeostasis in patients with MDS due to iron overload (IOL) plays a crucial role in oxidative stress activity in MDS. The interaction between hepcidin and ferroportin is fundamental for maintaining systemic iron homeostasis, as it regulates iron absorption and distribution. Changes in hepcidin levels can significantly affect iron export and plasma concentrations, particularly under conditions of iron deficiency or hemorrhage, as reviewed by Nemeth and Ganz [96] and further discussed by Roemhild et al. [97]. Hepcidin also has emerging importance in MDS pathophysiology and prognosis. Although high serum hepcidin levels are associated with MDS, they are not significantly different from those in healthy individuals; instead, the association appears to be influenced by MDS subtype, transfusion history, and genetic background [98].

Iron overload in hematological disorders typically arises from chronic blood transfusion and increased gastrointestinal absorption. Managing excess iron is critical, and iron chelation therapy (ICT) remains the primary approach for preventing tissue damage and organ dysfunction, as explained by Fibach and Rachmilewitz [99]. Indeed, chronic transfusion therapy is the leading cause of iron overload in MDS patients, as described by Gattermann and Rachmilewitz [100]. ICT has demonstrated effectiveness in restoring iron balance, lowering serum ferritin, and reducing hepatic and cardiac iron stores, all of which contribute to decreased morbidity and mortality [101,102].

In MDS-RS, ineffective erythropoiesis suppresses hepcidin synthesis, exacerbating the iron overload. In this context, ICT improves survival outcomes in transfusion-dependent patients by lowering the levels of toxic iron species and is considered the most accessible and effective chelation strategy [103]. Specifically, Liu et al. [104] reported that deferasirox (DFX) has been shown to improve hemoglobin, platelet, and neutrophil counts, potentially reducing mortality by minimizing hemorrhagic and infectious complications.

Beyond its systemic benefits, deferasirox also reduces oxidative stress markers at the cellular level. In fact, Rodríguez-García et al. [105] observed increased protein carbonylation, a marker of oxidative damage, in MDS bone marrow samples. Deferasirox treatment significantly reduced these levels. Four proteins with high carbonylation were identified: cytoplasmic actin, zinc finger protein 846, 14-3-3 protein zeta/delta, and l-lactate dehydrogenase A chain. Additionally, p21 expression, which was upregulated in patients with MDS, was downregulated after DFX treatment, suggesting a mechanistic link to its therapeutic effects.

Iron overload in MDS is driven by both transfusion practices and ineffective erythropoiesis. ICT not only reduces non-transferrin-bound iron (NTBI) and labile plasma iron (LPI) but may also directly improve overall survival, as discussed by Steensma and Gattermann [106].

The impact of iron overload on oxidative stress and disease progression has been well established. Excess iron catalyzes the production of ROS, disrupting cellular functions and leading to organ toxicity, cytopenia, and AML progression. Kim and Leitch [107] described how the resulting oxidative ROS damage to macromolecules, signaling pathways, and genomic integrity contributes to the further impairment of hematopoiesis and promotes clonal evolution. Therefore, ICT should be a critical consideration for improving both disease-free and overall survival in MDS.

Other iron chelators such as deferiprone have also demonstrated antioxidant effects. In a previous study conducted by Merkel et al. [108], deferiprone reduced ROS levels by 58.6%, 33.3%, and 39.8% in red blood cells, polymorphonuclear leukocytes (PMNs), and platelets, respectively. Additional improvements include reductions in phosphatidylserine exposure and lipid peroxidation, along with increased reductions in glutathione levels.

Interestingly, patients who responded positively to ICT had higher baseline expression levels of PI-PLCβ1, Cyclin D3, and PKCα, suggesting that these molecules may serve as biomarkers of treatment efficacy. According to Cappellini et al. [109], the expression of this pathway was downregulated following DFX treatment; this provides insight into the relevant potential mechanisms of action.

Moreover, iron overload-induced oxidative stress affects the self-renewal, proliferation, and differentiation capacities of hematopoietic stem cells, which are critical for maintaining effective hematopoiesis. ROS are implicated in clonal expansion, hematologic improvement, and recovery after hematopoietic stem cell transplantation (HSCT), as proposed by Pilo and Angelucci [110]. At diagnosis, patients with iron overload exhibited higher levels of superoxide anions than both healthy controls and patients with MDS without IOL. Additionally, Ivars et al. [111] reported significant alterations in the antioxidant defense system in IOL patients, including dysregulation of the catalase, glutathione peroxidase, and glutathione levels (Figure 1).

### 5.4. Mitochondrial Dynamics and Ferroptosis in the Metabolic Landscape of Myelodysplastic Syndromes

Ferroptosis is a regulated, iron-dependent form of cell death driven by the accumulation of lipid peroxides. It is distinct from apoptosis and is characterized by oxidative damage to cellular membranes, resulting from impaired antioxidant defenses, particularly glutathione peroxidase 4 (GPX4). This form of cell death has gained increasing relevance in cancer biology, especially in conditions involving metabolic reprogramming and iron overload, such as in myelodysplastic syndromes (MDS) [112].

In this context, ferroptosis is emerging as a novel therapeutic strategy, particularly for hematological malignancies that are resistant to conventional chemotherapy. These malignancies exhibit unique metabolic traits, including increased iron uptake and storage, which may enhance their sensitivity to ferroptotic triggers. As such, inducing ferroptosis represents a promising direction for the development of targeted therapies, provided that the molecular mechanisms of sensitivity and resistance are better understood [113].

Mitochondria play a central role in the regulation of ferroptosis, given their involvement in iron metabolism and reactive oxygen species (ROS) generation. Mitochondrial dynamics and interactions with other organelles also contribute significantly to the regulation of this death pathway [114]. In myeloid malignancies, including MDS, alterations in mitochondrial morphology—particularly excessive mitochondrial fragmentation—are associated with clonal evolution and disease progression. Notably, inhibition of the fission protein DRP1 has been shown to attenuate inflammatory signaling and improve survival in MDS preclinical models, highlighting a potential therapeutic avenue [115].

Moreover, metabolic rewiring in malignant cells, such as enhanced glycolysis, leads to a reorganization of mitochondrial tricarboxylic acid (TCA) cycle enzymes. This creates metabolic heterogeneity and alternative pathways that sustain increased cell proliferation. The Warburg effect, an “emergent property” of multiple integrated metabolic changes, further illustrates how altered energy metabolism supports tumor growth and contributes to the biology of leukemic transformation [116].

In normal hematopoietic stem cells (HSCs), anaerobic glycolysis is the predominant source of energy, and a high mitochondrial membrane potential is essential for maintaining quiescence and function. ROS regulation is critical in this setting, as oxidative stress impacts chromatin remodeling and cell fate decisions. Furthermore, nutrient pathways, including amino acid and fatty acid metabolism, contribute to acetyl-CoA production, which fuels the TCA cycle, despite the limited direct connection between glycolysis and mitochondrial respiration in HSCs [117].

Mitochondrial quality control (MQC) mechanisms are essential for preserving metabolic homeostasis in HSCs. Processes such as mitophagy remove dysfunctional mitochondria, thereby limiting ROS accumulation and preventing premature cellular aging. In parallel, mitochondrial biogenesis and intercellular mitochondrial transfer are crucial for maintaining energy balance and compensating for metabolic stress [118].

In fact, mitochondrial transfer has emerged as a therapeutic approach capable of restoring bioenergetic function by replacing damaged mitochondria with functional ones. This process plays a pivotal role not only in tissue repair and regeneration, but also in the pathogenesis, progression, and treatment resistance of hematologic malignancies, including AML and ALL [119].

Importantly, the oxidative stress and mitochondrial dynamics differ between low-risk and higher-risk MDS, with clear biological and clinical implications. Low-risk patients typically present with elevated basal ROS levels and reduced mitochondrial membrane potential (Δψ_m_), a profile that correlates with increased apoptosis and ineffective hematopoiesis1. In contrast, intermediate- and high-risk groups exhibit higher Δψ_m_ and enhanced OXPHOS capacity, features that support clonal proliferation and contribute to leukemic evolution [82]. Clinically, this divergence suggests that low-risk MDS may benefit from interventions targeting ROS accumulation or promoting ferroptosis, while high-risk MDS may be more vulnerable to therapies that disrupt mitochondrial potential or metabolic flexibility, offering a risk-adapted strategy to improve patient outcomes.

### 5.5. DNA Methylation and Epigenetic Regulation in Myelodysplastic Syndromes: Biomarkers and Functional Implications

Building on the connection between oxidative stress and disease mechanisms, epigenetic regulation, particularly DNA methylation, has emerged as a crucial factor in MDS pathophysiology. Changes in DNA methylation are considered potential biomarkers for diagnosis, prognosis, treatment response, and early toxicity prediction in both AML and MDS. As described by Dexheimer et al. [120], these alterations occur early during disease progression and can forecast myeloid or lymphoid transformation as well as therapeutic outcomes. Methylation profiling is currently being utilized for patient stratification, disease monitoring, and treatment optimization.

Furthermore, Gonçalves and colleagues [121] demonstrated a positive correlation between oxidative stress and DNA methylation levels in MDS patients. Higher global methylation was associated with poorer overall survival, highlighting the clinical significance of redox–epigenetic interactions. Specific oxidative stress markers—such as elevated peroxide levels and an increased peroxide/total antioxidant status (TAS) ratio—were found to be more prevalent in patients exhibiting promoter methylation of tumor suppressor genes, reinforcing a biochemical link between redox imbalance and epigenetic dysregulation [121]. In particular ROS were shown to correlate with the methylation status of the P15 and P16 gene promoters: patients with methylated genes had higher ROS levels, and parameters such as the peroxide/GSH ratio were proposed as potential discriminators between methylated and unmethylated gene profiles [122].

In addition to DNA methylation, transcriptional regulators play a pivotal role in MDS pathogenesis. One such factor is the myeloid lineage master regulator CCAAT enhancer-binding protein alpha (C/EBPα), encoded by the *CEBPA* gene. This transcription factor is essential for granulocytic lineage commitment and cell-cycle arrest. Its functional inhibition represents a leukemogenic event, as shown by Pulikkan, Tenen, and Behre [123], and restoration of C/EBPα levels can trigger granulocytic differentiation and prolong survival in leukemia models.

In MDS, CEBPA promoter methylation is significantly more frequent than in healthy controls, indicating its importance in disease development. As reported by Wen et al. [124], CEBPA mutations are rare and tend to occur mutually exclusively with respect to promoter methylation events. Despite its frequency, CEBPA methylation does not correlate strongly with transcriptional repression or patient outcome. Nevertheless, oxidative stress is hypothesized to contribute to aberrant CEBPA methylation, linking redox imbalance to defective granulocytic differentiation and disease progression in MDS.

Another key player in the epigenetic landscape is TET2, an Fe^2^⁺- and α-ketoglutarate-dependent dioxygenase involved in active DNA demethylation. As discussed by Petzer et al. [125], TET2 mutations are common in MDS and have been shown to impair natural killer (NK) cell function by altering methylation patterns, leading to reduced expression of cytotoxic effector molecules such as KIRs, perforin, and TNF-α. These immune defects contribute to immune evasion and disease persistence. Importantly, Boy et al. [126] reported that treatment with hypomethylating agents (HMAs) like azacitidine can restore KIR expression and partially normalize NK cell function. Moreover, oxidative stress has been found to impair TET2 enzymatic activity, further reinforcing the connection between redox imbalance and the epigenetic alterations commonly observed in MDS.

### 5.6. Oxidative Stress in MDS and the Treatment Options Available

Oxidative stress in MDS can also serve as a valuable therapeutic target. In a recent study by Harris et al. [127], reduced expression of RPL22, a ribosomal protein, was found to be frequent in both MDS and AML patients and correlated with poorer clinical outcomes. The authors demonstrated that RPL22 deficiency enhances the self-renewal capacity of HSCs and impairs their differentiation by upregulating ALOX12, a key regulator of fatty acid oxidation (FAO). The resulting elevation of FAO supports leukemic cell survival, identifying a potential therapeutic vulnerability in RPL22-deficient MDS and AML.

Another promising axis involves arginine metabolism. Gonzalez-Menendez and colleagues [128] demonstrated that arginine uptake and its subsequent conversion to spermidine are crucial for the hypusination of eukaryotic initiation factor 5A (eIF5A), which plays a pivotal role in human erythroid differentiation. Notably, inhibition of hypusine synthesis selectively disrupts erythropoiesis while sparing myeloid differentiation, suggesting a lineage-specific function for eIF5A. These findings also highlight the regulatory role of ribosomal proteins in coupling the mitochondrial metabolism with red blood cell development.

Among the pharmacological agents, decitabine (DAC) is known to induce cell-cycle arrest and apoptosis in leukemia cells, triggering delayed ROS accumulation. This effect is independent of p53 and requires deoxycytidine kinase (DCK) activity [104]. DAC also upregulates ROS-generating enzymes, particularly NADPH oxidase isoforms, and increases Nox4 protein expression in an ATM-dependent manner, highlighting its link to DNA damage signaling and redox modulation [129].

Clinical evidence supports the therapeutic relevance of decitabine. In a multicenter phase II trial reported by Lübbert and colleagues [130], DAC was administered as a first-line treatment to elderly AML patients unfit for intensive chemotherapy. The study achieved a combined complete and partial remission rate of 26%, with a median overall survival of 5.5 months and a one-year survival rate of 28%, confirming its tolerability despite frequent myelosuppression.

Beyond decitabine, Montes et al. [131] explored the redox-modulatory effects of 5-azacitidine in patients with MDS. Their findings demonstrated that treatment with 5-azacitidine led to elevated oxidative stress markers, particularly increased ROS levels, without significantly affecting inflammatory parameters. Interestingly, prior to treatment, MDS patients showed a more favorable redox profile compared to healthy controls, as reflected by the lower ratio of oxidized glutathione (GSSG) to reduced glutathione (GSH), reduced lipid peroxidation, and higher catalase (CAT) activity.

In addition, upregulation of the HIF-1α signaling pathway has been associated with improved responses to 5-azacitidine in patients with high-risk MDS and AML. Elevated HIF-1α expression serves as an independent predictor of treatment response and correlates with better overall survival and a delayed progression to AML [132].

Vitamin C has also emerged as a promising redox-based therapeutic candidate. As reviewed by Gillberg et al. [133], vitamin C deficiency is commonly observed in cancer and may contribute to malignant transformation by impairing epigenetic regulation.

Notably, vitamin C levels are significantly lower in AML patients compared to healthy donors and show an inverse correlation with peripheral blast counts. These levels increase upon complete remission and decline further in cases of resistant disease. Additionally, the expression of vitamin C transporters is reduced in AML patients, although vitamin C concentration does not appear to predict survival or relapse [134].

At physiological levels, vitamin C synergizes with hypomethylating agents (HMAs) to enhance DNA demethylation via activation of TET and JHDM enzymes. At pharmacological doses, however, it promotes ROS production, selectively inducing cytotoxicity in neoplastic cells. In support of this dual mechanism, Cimmino et al. [135] demonstrated that vitamin C can mimic TET2 reactivation, reverse aberrant self-renewal in pre-leukemic stem cells, and restore normal DNA methylation dynamics, highlighting its therapeutic potential in MDS.

Furthermore, vitamin C has been shown to decrease cell proliferation and viability in a concentration-, time-, and administration-dependent manner, inducing apoptosis in F36P cells, a human erythroleukemia cell line commonly used to study erythroid differentiation and growth-factor responses. Its pro-oxidant effects, particularly the generation of superoxide and mitochondrial membrane depolarization, represent key mechanisms underlying its cytotoxic activity. Together, these findings support the exploration of vitamin C as a novel therapeutic strategy for MDS [136].

Natural compounds also contribute to the development of redox-targeted therapies. Among them, Withaferin A (WFA), a steroidal lactone derived from Withania somnifera (Ashwagandha), has shown antioxidant, anti-inflammatory, and anticancer properties through modulation of multiple oncogenic pathways [137]. A 2017 study demonstrated that WFA exerts selective cytotoxicity against MDS cells, particularly the MDS-L cell line, while sparing normal bone marrow cells. This selective effect is primarily mediated by the activation of the ROS-dependent JNK/AP-1 signaling pathway, ultimately leading to apoptosis. Thus, ROS generation is central to the therapeutic action of WFA in MDS [138].

In addition to its effects in MDS, WFA has demonstrated growth-suppressive activity in leukemia and lymphoma cell lines by inducing cell-cycle arrest at the G2/M phase. Although only partial apoptosis is observed in these models, its cytotoxicity can be significantly enhanced by combining WFA with autophagy inhibitors such as chloroquine. This combination strategy results in amplified apoptosis and tumor growth suppression, suggesting a promising redox-modulating approach in hematologic malignancies [139].

Finally, nicotinamide riboside (NR), another compound of interest, reduces the mitochondrial activity in HSCs by promoting mitochondrial clearance. This enhances asymmetric stem cell division, expands the progenitor pool, and improves survival and blood recovery after irradiation and transplantation. These findings link mitochondrial stress, mitophagy, and stem cell fate, highlighting the potential of NR in the treatment of hematological failure [140].

## 6. Future Perspectives in Oxidative Stress Research: The Role of Artificial Intelligence in Antioxidant Discovery in Myeloid Malignancies

Among these compounds, quercetin stands out for its multifunctional biological properties, including its antioxidant, antiproliferative, apoptotic, and autophagic effects. It has been shown to reduce the growth of various cancer cell lines, including AML and MDS cells. Through its multitarget activity, quercetin has demonstrated potential as a therapeutic agent in the treatment of AML and MDS by inducing cancer cell death [141].

Emerging technologies, particularly artificial intelligence (AI) and machine learning (ML), also contribute to our understanding of oxidative stress in hematological malignancies. A Korean study developed an ML model that accurately predicted oxidative stress levels in healthy individuals, using receiver operating characteristic (ROC) curves, and achieving high sensitivity and specificity [142].

AI and ML tools are being increasingly applied to evaluate antioxidant capacity by correlating the chemical structure (via molecular descriptors) with in vitro antioxidant activity. These models enhance compound screening efficiency, while supporting adherence to ethical research standards [143]. External validation using data from the BATMAN database confirmed the reliability of Random Forest (RF) and Support Vector Machines (SVM) models in predicting antioxidant potential [144].

In a recent GPT-based literature analysis, 1416 pro-oxidant and 1102 antioxidant compounds were identified and classified into chemical categories, such as flavonoids, thiols, pharmaceuticals, pesticides, and metals [145].

One promising area where these computational strategies are being applied is drug repurposing—the search for new therapeutic uses for existing or experimental compounds. This approach offers the advantage of reducing both the time and costs associated with drug development. Nevertheless, the field faces challenges such as legal uncertainties, intellectual property concerns, and regulatory barriers. Proposed strategies aim to address these obstacles and fully leverage the potential of repurposed drugs [146]. ML and AI enhance this process by offering cost-effective methods for identifying new applications of approved drugs. These supervised techniques often rely on publicly available databases, although their effectiveness is limited by the scarcity of clinical patient data and an overreliance on genomics-based information. Functional assays on patient-derived cancer cells can provide additional insight, complement in silico predictions, and help to mitigate the financial and developmental risks inherent in drug discovery [147].

Computational tools, including ML and deep learning models, are also being applied to uncover novel drug-target relationships in oncology. Among the therapeutic strategies under investigation, microRNA modulation has emerged as a key approach. Advances in Next Generation Sequencing (NGS) have significantly expanded our understanding of cancer biology, providing the high-quality data essential for the accuracy of ML-based drug repurposing models. Despite these advances, the clinical potential of drug repurposing in oncology remains largely untapped [148].

Finally, AI and ML are also revolutionizing traditional drug discovery by addressing longstanding challenges such as high development costs and lengthy timelines. Sophisticated AI frameworks, including deep neural networks (DNNs), convolutional neural networks (CNNs), and deep reinforcement learning (DRL), are being used to identify drug targets and enhance delivery systems. Personalized treatment strategies are also being refined through models like CURATE.AI and the Hartung algorithm, which aim to optimize therapeutic efficacy and predict toxicological risks [149].

## 7. Conclusions

Myelodysplastic syndromes represent a heterogeneous group of clonal hematopoietic disorders driven by complex genetic, epigenetic, and metabolic alterations. Among the emerging contributors to the pathophysiology of MDS, oxidative stress has gained increasing recognition with respect to its roles in promoting genomic instability, disrupting hematopoietic stem cell function, and influencing disease progression. Elevated ROS levels, often exacerbated by iron overload, mitochondrial dysfunction, and impaired antioxidant defenses, are associated with the key clinical features of MDS, including ineffective erythropoiesis and increased leukemic transformation risk. Therapeutically, redox-modulating strategies, such as iron chelation, antioxidant supplementation, and agents that influence ROS-mediated signaling pathways (e.g., decitabine, vitamin C, and Withaferin A), offer promising adjuncts to conventional treatment regimens. These approaches may reduce oxidative damage, restore redox balance, and enhance hematologic responses, particularly in transfusion-dependent or lower-risk MDS patients. From a clinical perspective, the integration of oxidative stress markers (e.g., malondialdehyde, protein carbonylation, glutathione levels, and ROS/GSH ratios) into existing risk stratification models could refine prognostic assessments and personalize treatment choices. Measuring these parameters routinely may help identify the patients more likely to benefit from iron chelation or redox-targeted therapies. Future research should focus on validating oxidative stress biomarkers in prospective clinical trials and exploring their predictive value for treatment response and disease evolution. In parallel, further mechanistic studies are needed to clarify the interactions between redox imbalance, clonal hematopoiesis, immune dysfunction, and epigenetic deregulation. The application of artificial intelligence and machine learning to large genomic and biochemical datasets may facilitate the discovery of new therapeutic targets and support drug repurposing strategies. Ultimately, a more comprehensive integration of redox biology into MDS management could pave the way for precision medicine approaches that optimize outcomes based on molecular and metabolic patient profiles.

## Figures and Tables

**Figure 1 ijms-26-06415-f001:**
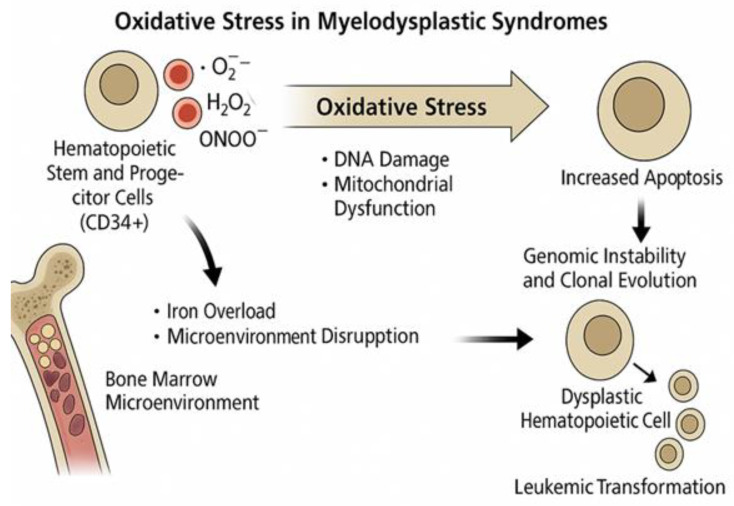
Effects of iron overload in MDS.

**Table 1 ijms-26-06415-t001:** Key mechanisms linking oxidative stress to MDS.

Mechanism	Molecular Drivers	Cellular Impact	Therapeutic Strategies
DNA Damage	ROS, RNS	Genomic instability, mutations, apoptosis	Antioxidants, DNA repair enhancers
Mitochondrial Dysfunction	Iron overload, mtDNA mutations	Altered energy metabolism, increased ROS	Iron chelators (deferasirox), mitochondrial-targeted therapies
Epigenetic Alterations	TET2, DNMT3A, CEBPA methylation	Aberrant gene expression, impaired differentiation	HMAs, vitamin C (TET cofactor)
Redox Imbalance	Low GSH, high NOX activity	Sustained oxidative stress, ineffective hematopoiesis	GSH donors, NOX inhibitors
Immune Dysfunction	NK cells, inflammatory cytokines	Impaired immune surveillance	Immunomodulators, azacitidine

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
