# Peer review of "Oxidative Stress and Mitochondrial Dysfunction in Myelodysplastic Syndrome: Roles in Development, Diagnosis, Prognosis, and Treatment"

_ijms, 2025, doi:10.3390/ijms26136415_

Round 1

Reviewer 1 Report

Comments and Suggestions for Authors

Dear Authors,

The manuscript entitles “Oxidative Stress and Mitochondrial Dysfunction in 2 Myelodysplastic Syndrome: Role in Development, Diagnosis, 3 Prognosis, and Treatment” presents novelty and scientific advance

The article comprehensively addresses the clinical, molecular, and therapeutic aspects of myelodysplastic syndrome (MDS), with a special focus on oxidative stress. Recent references (up to 2024) are included, demonstrating an up-to-date literature review. The inclusion of artificial intelligence and machine learning in MDS research is innovative and relevant.

Minor revision is my piece of advice.

Suggestions:

The terminology consistency: Make sure you consistently use terms like “oxidative stress,” “ROS,” “redox imbalance,” etc., to avoid confusion.

In the conflicts of interest section please remove “decare” and write “declare”

References: Although numbered, some in-text citations are not well integrated into the narrative flow. Phrases such as "According to Smith et al. [15]..." are recommended to improve readability.

Tables and Figures: Table 1 and Figure 1 are cited but could benefit from better visual presentation (alignment, table formatting, clearer legends).

The conclusion is sound but could benefit from a final section that summarizes clinical recommendations or future lines of research more concretely.

Best Regards

Author Response

Dear Reviewer many thanks. I revised the paper accordingly to your indications.

Reviewer 2 Report

Comments and Suggestions for Authors

This manuscript presents a review of the role of oxidative stress and mitochondrial dysfunction in the pathobiology of myelodysplastic syndromes (MDS). The authors integrate recent advances in molecular biology, genetics, and therapeutic strategies to emphasize the importance of oxidative stress in disease progression, diagnosis, and treatment. While the manuscript may serve as a useful resource for researchers and clinicians in the field of hematologic malignancies, I do not recommend it for publication in the current journal in its present form, for several key reasons:

  1. Lack of mechanistic insight into oxygen regulation: Although oxidative stress is highlighted, the manuscript does not adequately address how oxygen availability or regulation is altered in MDS and acute myeloid leukemia (AML). Beyond listing existing research findings, the authors should discuss the mechanisms by which oxidative stress contributes to the progression from MDS to AML.

  2. Conceptual vagueness of oxidative stress: Oxidative stress is a broad and sometimes ambiguous concept. The review would benefit from a more nuanced discussion of how different levels of reactive oxygen species (ROS) impact disease biology. For instance, while high levels of ROS can drive genomic instability and leukemogenesis, low levels of ROS such as Hâ‚‚Oâ‚‚ may act as signaling molecules that promote tumor stem cell maintenance and expansion. These dual roles should be clearly addressed, especially in the context of the uncertain efficacy of antioxidant-based therapies in these diseases.

  3. Incomplete discussion of mitochondrial heterogeneity and metabolism: Given that the manuscript also touches on mitochondrial dysfunction, it should consider the evolving heterogeneity of mitochondria during leukemic transformation. Specifically, the review should explore how mitochondrial functional states contribute to oxidative stress, and how these are influenced by altered glucose and lipid metabolism as well as ferroptosis. Additionally, examining how oxidative stress mechanisms vary across different MDS risk groups would provide important biological and clinical insights.

Author Response

(The authors gave the same response as above.)

Reviewer 3 Report

Comments and Suggestions for Authors

This manuscript entitled “Oxidative Stress and Mitochondrial Dysfunction in Myelodysplastic Syndrome: Role in Development, Diagnosis, Prognosis, and Treatment” provide a good description of MDS and recent advances in oxidative stress and mitochondrial dysfunction in MDS. It also provides some useful clinical information. Having said that, the authors should shorten the general description of MDS and focus more on how oxidative stress and mitochondrial dysfunction induce affect HSPC differentiation and induce dysplastic myelopoiesis. For example, it is better to include some recent studies on the differential roles of oxidative stress and mitochondrial dysfunction in MDS vs. bone marrow failure syndromes. The following are some minor concerns/comments.

Lines 70-71: Regrading the sentence of “Consequently, molecular profiling using NGS has become the cornerstone of diagnosis and therapeutic planning for MDS [10]”, I would suggest changing to “one of the cornerstones” because clonal hematopoiesis (CH) mutations are indeed a common phenomenon observed in aging individuals and most of them do not develop MDS. Epigenetic and functional changes as well as morphological dysplasia are also the cornerstones in diagnosis and treatment of MDS.

Lines 74-75:  “…is often described through the "Yin-Yang theory" of immune activation and suppression, and is characterized by immune hyperactivity and increased apoptosis in some low-risk/low-grade MDS…[11].” The words of low-risk MDS (LR-MDS) should be add here because such immune changes are seen in some LR-MDS, but typically not in high-risk MDS cases.

Lines 86-176: The sections of 2. Risk Stratification and Treatment Options and 3. New Treatment Approaches” are much redundant and overly broad. Furthermore, there have been numerous review articles published on these topics. The authors should shorten these sections as much as possible and focus more on the recent advances in oxidative stress and mitochondrial dysfunction in MDS.

Line 280: It should be spelled out here as D-2-hydroxyglutarate (D-2HG); Prolyl hydroxylases (PHDs).

Lines 317-24: The authors should explain how NUP98-HOXD13 fusion increases DNA double-strand breaks and affects oxidative status and mitochondrial function.

Line 319: The word 39.8% is redundant. It should be deleted and changed to red blood cells, polymorphonuclear leukocytes (PMNs), and platelets, respectively.

Line 423: Please spell out it as Total Antioxidant Status (TAS).

Line2 481-482: Need to spell out the GSSG/GSH/ ratio, the ratio of oxidized glutathione (GSSG) to reduced glutathione (GSH).  Of note is that the reference (no 106), in my opinion, is an interesting and informative study because 5-AZA is widely used to treat MDS and was reportedly to increase oxidative stress, which seems to have a different mechanism of action (MOA) that seems contradictory to the MOC of Withaferin A. The authors may need to talk about more about these studies including 5-AZA, decitabine, vitamin C, and Withaferin A in MDS.

Author Response

(The authors gave the same response as above.)
